# Satisfaction Paradoxes in Health Behaviors: Contrasting Patterns across Weight, Physical Activity and Dietary Habits

**DOI:** 10.3390/nu16142246

**Published:** 2024-07-12

**Authors:** Mohammed A. Muaddi, Anwar M. Makeen, Ibrahim M. Gosadi, Mohammad A. Jareebi, Abdullah A. Alharbi, Ahmed A. Bahri, Majed A. Ryani, Mohamed Salih Mahfouz, Osama Albasheer, Suhaila A. Ali, Abdulmajeed A. Arishi, Fatima A. Alsam, Ahmad Y. Alqassim

**Affiliations:** 1Family and Community Medicine Department, Faculty of Medicine, Jazan University, Jazan 45142, Saudi Arabia; mothman@jazanu.edu.sa (M.A.M.); amakeen@jazanu.edu.sa (A.M.M.); gossady@hotmail.com (I.M.G.); mjareebi@jazanu.edu.sa (M.A.J.); aaalharbi@jazanu.edu.sa (A.A.A.); dr.bahri2010@gmail.com (A.A.B.); majedryani@gmail.com (M.A.R.); mm.mahfouz@gmail.com (M.S.M.); oalbasheer@jazanu.edu.sa (O.A.); zogla1974@gmail.com (S.A.A.); 2Faculty of Medicine, Jazan University, Jazan 45142, Saudi Arabia

**Keywords:** obesity, physical activity, diet, lifestyle, satisfaction, body mass index, university students, Saudi Arabia, health behaviors, dietary habits

## Abstract

(1) Background: Obesity, a poor diet, and inactivity are major health issues among Saudi youth. However, satisfaction with unhealthy lifestyles could impede change. This study assessed lifestyle factors and related satisfaction among Saudi university students. (2) Methods: In this cross-sectional study, 1957 students at Jazan University completed surveys on demographics, physical activity, dietary habits, and 10-point satisfaction scales for weight, activity, and diet. Chi-squared tests and logistic regression were used to analyze the associations between behaviors and satisfaction. (3) Results: Overweight/obesity prevalence was 25.45%, and only 26.67% of the subjects met activity guidelines. Many of them exhibited poor dietary habits. Despite unhealthy behaviors, some expressed high satisfaction, especially regarding their diets. Subjects with a normal BMI had the highest weight satisfaction. Activity satisfaction increased with higher activity levels. Dietary satisfaction was minimally impacted by healthfulness. Males and higher incomes were correlated with greater satisfaction. (4) Conclusions: A concerning paradox exists between unhealthy lifestyles and satisfaction among Saudi university students, particularly regarding their diets. Multicomponent interventions informed by behavior change theories and employing motivational techniques are urgently needed to address this disconnect and facilitate positive behavior change.

## 1. Introduction

Obesity, a poor diet, and physical inactivity have emerged as major public health challenges among young adults worldwide [1]. Compared to global averages, Saudi Arabia (SA) experiences disproportionately high obesity rates (35% versus 13% worldwide) [2] and obesity-related mortality (18% of deaths versus 8% globally) [3]. This elevated burden is likely fueled by a high prevalence of sedentary lifestyles and poor dietary habits, as noted across various studies of young Saudi adults [4,5,6]. These unhealthy lifestyles significantly increase risks for non-communicable diseases like diabetes, cardiovascular disease, and cancer [7]. The resulting morbidity not only compromises people’s quality of life on an individual level but also threatens the health of future populations, healthcare capacity, workforce productivity, and economic progress across SA [8,9].

In this study, satisfaction refers to an individual’s overall contentment and positive assessment of their current lifestyle behaviors, specifically their body weight status, level of physical activity, and dietary habits. A growing body of research highlights concerning rates of obesity, inactivity, and poor diets among university students in SA and neighboring countries [10,11,12,13]. However, associations between health behaviors and related satisfaction remain unclear. Some studies reveal coexistence of obesity and related risks with high levels of satisfaction that can be referred to as a “risk paradox”. In contrast, other evidence indicates that lower satisfaction is related to inactivity or being overweight [14,15,16]. The promotion of nutritional changes is a major challenge compared to increasing exercise, with taste preferences, ingrained habits, and low self-efficacy hampering dietary improvements despite there being various incentives to eat healthier [17,18,19].

Given these conflicting findings and the concept of the “satisfaction paradox”, where individuals exhibit risks like obesity yet report high satisfaction, the current study aimed to elucidate relationships between lifestyle factors and satisfaction levels among Saudi university students. We hypothesized that students exhibiting obesity, physical inactivity, and poor dietary habits would report lower satisfaction compared to students with healthier lifestyles. Recognizing these risks and perceptions will inform tailored interventions to mitigate obesity and chronic disease burdens impacting Saudi youth.

To promote health, the World Health Organization (WHO) recommends regular physical activity and consuming a diet rich in fruits, vegetables, and whole grains while limiting the intake of fat, sugar, and salt [20]. The American Heart Association recommends a minimum of 150 min of moderate exercise per week for optimal health [21]. However, both physical inactivity and poor dietary habits are highly prevalent among young Saudi adults, constituting major public health concerns [10,22]. Studies have found low rates of meeting recommended physical activity guidelines among Saudi adults, with only 15.0–33.2% engaging in 150 min or more of moderate activity per week [23,24]. Regarding diet, frequent fast food consumption, high intake of sugar-sweetened beverages, and low fruit/vegetable intake characterize a typical pattern seen in this population [10]. For instance, only 3.27% of Saudi university students meet the daily fruit/vegetable recommendations [25], while a study of female university students found that 97% of them consume fast food daily [26]. More broadly, results from a nationally representative survey indicate that dietary guideline recommendations are met by only 5.2% and 7.5% of Saudis for fruits and vegetables, respectively [10].

In an effort to address the significant health and societal consequences associated with obesity, the government of SA is enacting a comprehensive set of policies as part of its Vision 2030 initiative, which is aimed at promoting a healthier lifestyle. The Vision 2030 plan prioritizes the promotion of healthy lifestyles and the prevention of obesity as its key objectives. This forward-thinking policy framework aims to foster environments, resources, and messages that encourage increased physical activity and improved dietary behaviors across the Saudi population [27].

While previous studies have documented a high prevalence of sedentary lifestyles, unhealthy eating patterns, and elevated obesity levels among the Saudi adult population, few have examined their own satisfaction regarding these behaviors. This represents a key gap within our knowledge as satisfaction with potentially unhealthy lifestyle choices could constitute a barrier to positive behavior changes [28,29]. Evidence on whether such a disconnect between health behaviors and perceptions exists within Saudi adults is limited.

Therefore, the aim of this study was to assess physical activity patterns, dietary habits, weight status, and related satisfaction levels among students at Jazan University. Recognizing lifestyle risks and perceptions is vital for mitigating immediate and long-term obesity and chronic disease burdens impacting the well-being and productivity of Saudi youth, with university settings offering a valuable opportunity for health promotion initiatives.

## 2. Materials and Methods

### 2.1. Study Design and Setting 

This work is part of a larger project carried out in May and June 2023 to assess the health needs of Jazan University’s faculty members, administrative staff, and undergraduate students [30,31]. This research employed a cross-sectional study design that was conducted among Jazan University affiliates. Jazan University is a leading higher education institution in southwest Saudi Arabia. The university was established in 2005 and is populated by more than 50,000 students across 23 colleges, with 21 of them offering bachelor’s degree programs and 1 offering diploma programs.

### 2.2. Population and Sampling Design

This study targeted Jazan University’s undergraduate students aged 18 years and older who were registered for the academic year 2022/2023. The sample size was calculated to be 2291 students based on a sample size formula for estimating a single proportion [32]. The parameters used in the calculation involved a 95% confidence interval, an expected proportion (p) presumed as 50% (representing the proportion generating the largest sample size since this study measured several lifestyle characteristics), and a margin of error (d) set at 2% to reflect the desired precision. Convenience sampling was used to recruit study participants from various university colleges. To ensure representative sampling, we selected a proportional number of students from the chosen colleges.

### 2.3. Data Collection and Study Tool 

The data were gathered through personal interviews conducted by trained medical students using a standardized questionnaire. Recruitment involved identification and direct approach of targeted students on university campus. Students who agreed to participate completed the interview.

The questionnaire captured information on demographics, lifestyle factors, dietary habits, physical activity, tobacco and khat use, nutrition, weight, and height. Assessed variables aligned with Saudi guidelines on the prevention and management of obesity [33]. Self-reported physical activity was evaluated by asking if participants met the recommended 150 min of physical activity per week threshold. Dietary behaviors were assessed through questions on adherence to healthy eating recommendations, specifically, consumption of whole-grain products, fruits and vegetables, low-fat meats, and low-fat products, as well as avoidance of high sugar foods. The students were also asked to rate their overall satisfaction with their lifestyles. Detailed information about the questionnaire, including its validity and reliability, was published elsewhere [31].

### 2.4. Study Variables and Measures 

The main study outcomes were satisfaction levels related to body weight, physical activity, and eating behaviors among university students in Jazan. Independent variables included demographic characteristics, eating behaviors, physical activity, body mass index (BMI), smoking, and khat chewing. 

BMI was calculated based on measured height and weight using the standard formula of weight in kilograms divided by height in meters squared. The BMI was categorized according to World Health Organization (WHO), which suggests that individuals with BMI < 16.0 kg/m^2^ are underweight, BMI = 18.5–24.9 kg/m^2^ are normal weight, BMI = 25.0–30.0 kg/m^2^ are overweight, and BMI ≥ 30.0 kg/m^2^ are obese [34]. 

Satisfaction for each domain (weight, activity, diet) was measured on a 10-point scale, with a score of 5 or above defined as satisfaction and below 5 as dissatisfaction.

### 2.5. Statistical Analysis 

The data analysis was conducted using R software (version 4.2.3). The analysis included both descriptive statistics and inferential statistics. Continuous numerical variables with a normal distribution were reported as means and standard deviations (SDs), while variables with skewed distribution were presented as medians and interquartile range (IQR). The Mann–Whitney U test was used to assess the relationship between eating behavior variables and satisfaction.

Median satisfaction and IQRs were compared across BMI categories, physical activity levels, and adherence to healthy eating recommendations. For binary analyses, a cutoff of 5 was used to categorize satisfaction levels. Participants with scores of 5 or higher were classified as satisfied, while those below 5 were classified as dissatisfied.

Logistic regression analysis was also used to estimate the adjusted odds ratio (AOR) for selected covariates with the outcome variables. Students’ satisfaction with their weight, physical activity levels, and dietary behaviors were examined as outcome variables. A p-value of less than 5% was considered statistically significant. Additionally, Microsoft Excel was utilized to graph and visualize the satisfaction levels for the different lifestyles studied.

To assess the robustness of the findings and the impact of the chosen satisfaction thresholds on the observed paradox, analyses were conducted using alternative cutoff values (6, 7, and 8) on the 10-point satisfaction scale. The results of the sensitivity analysis using the alternative cutoff values are presented in the Appendix A. Additionally, stratified analyses were performed to examine the consistency of the results across different subgroups, such as gender, age, and socioeconomic status. 

## 3. Results

This cross-sectional study included 1957 Saudi university students with a mean age of 21 years. As shown in Table 1, the sample contained a nearly equal proportion of males and females. Height and weight differed by gender, with greater values being seen among males compared to females, resulting in higher average BMIs. Participants’ places of residence were also evenly split between rural and urban areas. Over half of the participants had relatively low family incomes, were single, and lived in owned housing. Overall, participants’ rates of smoking and khat use were low. Regarding BMI status, over half of the participants were normal weight, while around 25% were overweight/obese. Many participants were insufficiently physically active, with only about 27% meeting the guidelines. Around half of the participants exhibited some healthy eating habits. The proportion who reported engaging in some of the outlined healthy dietary habits ranged from about 26% to 54%.

Table 2 displays the median and IQR of satisfaction levels across BMI categories, physical activity levels, and selected eating behaviors. Satisfaction levels demonstrated a reverse J-shaped pattern across BMI categories, with the highest satisfaction among normal weight and the lowest satisfaction among obese individuals (Figure 1). For physical activity, satisfaction showed a dose–response relationship (Figure 2), with the highest satisfaction reported among participants achieving the recommended 150 min of or more of physical activity per week. Those engaging in less than 150 min of physical activity per week displayed lower satisfaction, while participants reporting no activity had the lowest satisfaction. Minimal differences were observed in satisfaction levels between individuals that adhered to healthy eating behaviors compared to those who did not (Figure 3). 

Table 3 shows the distribution of satisfaction levels for body weight, physical activity, and eating behaviors. Satisfaction was defined by a score of 5 or higher on a 10-point scale, while dissatisfaction was a score below 5. Satisfaction levels varied noticeably across different BMI categories, with the highest dissatisfaction being recorded among obese participants and the highest satisfaction among normal weight participants. For physical activity, individuals engaging in no physical activity predominantly expressed dissatisfaction, while those engaging in 150 min or more of physical activity per week demonstrated higher satisfaction. Minimal differences were observed in satisfaction levels among individuals that practiced healthy eating habits compared to those who did not. 

The multivariate analysis reveals the predictors associated with satisfaction levels regarding weight, physical activity, and eating behaviors. As demonstrated in Table 4, factors like older age, male gender, higher income, urban residence, normal BMI, meeting physical activity guidelines, and healthy eating habits were associated with greater satisfaction in one or more domains. Conversely, overweight/obesity tended to predict lower satisfaction.

## 4. Discussion

This study aimed to assess lifestyle choices and relevant satisfaction among university students in Jazan, Saudi Arabia. We hypothesized that students exhibiting obesity, physical inactivity, and poor dietary habits would report lower satisfaction levels compared to students with healthier lifestyles. However, the current findings revealed a more complex relationship. The findings provide valuable insights into the high prevalence of obesity, poor dietary habits, and physical inactivity among students. However, they also revealed a paradoxical relationship between dietary habits and perceived satisfaction. Individuals exhibited minimal differences in satisfaction levels regardless of their adherence to healthy eating guidelines. This finding is significantly different comparted to the more expected patterns seen for weight and physical activity satisfaction, where those with suboptimal BMIs or insufficient activity expressed lower satisfaction. The data point to a disconnect between dietary behaviors and perceived satisfaction, whereas satisfaction tended be more closely related with weight status and activity levels. A further exploration of this satisfaction/behavior discrepancy regarding people’s diets is necessary, as it may constitute a barrier to positive nutritional changes among students engaging in objectively unhealthy eating habits yet reporting satisfaction with their habits.

The observed prevalence of overweight/obesity (25.45%), no/low activity (73.33% not meeting guidelines), and the significant proportion of participants exhibiting suboptimal dietary habits aligned with the trends noted in other studies of young adults in Saudi Arabia and neighboring countries [10,11,12,13]. This highlights the profound impacts of nutrition transitions, technological advances reducing activity, and the adoption of Western lifestyles, which are associated with modernization and urbanization in the region [35,36]. The findings underscore the urgent need for comprehensive interventions to promote healthy dietary practices, active living, and weight management on university campuses to mitigate the risk of future disease among this vulnerable cohort [37].

Alarmingly, despite engaging in objectively unhealthy behaviors, some participants reported high levels of satisfaction related to their weight, diet, and physical activity. This “satisfaction-behavior paradox” suggests that many students may be in the precontemplation stage and not recognize the problematic nature of their habits [38]. The pronounced dietary paradox indicates that these students lack motivation to improve poor nutritional choices, representing a key barrier to positive change.

While most pronounced for diet, satisfaction/behavior disconnects existed to some degrees across domains. The dose–response relationship between physical activity, obesity, and satisfaction aligns with previous research, showing gratification from measurable improvements [14,15,16]. However, some still expressed satisfaction despite insufficient activity or unhealthy weight, highlighting the complexity of this relationship.

The weaker association of dietary habits with satisfaction highlights the unique challenges of promoting nutritional change, likely stemming from the difficulties of sacrificing tasty food for long-term health [17], ingrained habits [18], and lack of nutritional self-efficacy. Although motivated to eat healthier, many struggle to translate intentions into sustained action [19]. 

Executing a truly balanced diet requires people to consistently coordinate complex behaviors, which could account for why satisfaction is detached from objective dietary quality. These challenges appear to perpetuate the risk paradox between poor dietary choices and perceived satisfaction to a greater degree compared to activity and weight [39]. Though less pervasive, obstacles like body image perceptions, low fitness awareness, and environmental barriers may also contribute to inflated satisfaction ratings despite certain individuals having an unhealthy weight or being inactive [40]. 

These insights indicate a need for personalized, stage-based interventions to promote change among students exhibiting risk paradoxes. University initiatives should employ motivational interviewing to raise awareness and resolve the ambivalence regarding unhealthy behaviors. Facilitating incremental shifts, providing resources, and modifying environmental cues can help progress students through precontemplation. Qualitative research could inform motivational strategies tailored to young adults. Comprehensive interventions using personalized messaging and consciousness-raising activities can prompt readiness to change [41,42,43,44]. By meeting students with a gradual, compassionate approach, universities can address gaps and barriers at the precontemplation stage.

However, other studies have reported more predictable associations between healthy dietary practices and higher satisfaction levels [45,46,47], highlighting the complexity of this relationship. The stark contrast between the dietary habits’ paradox and the expected associations for weight and physical activity satisfaction observed in this study of young university students has important implications for health promotion efforts. Individuals that were satisfied with their dietary habits despite displaying poor adherence to nutritional guidelines may lack motivation for positive dietary changes. In less-educated or elderly populations, the paradox may manifest differently or be exacerbated by factors such as lower health literacy, cultural traditions, socioeconomic constraints, and age-related physiological changes [48,49,50]. Addressing this paradox requires multifaceted interventions that enhance people’s nutritional literacy, foster accurate self-evaluation, and empower individuals to make informed dietary choices that are aligned with evidence-based guidelines.

Several sociodemographic and behavioral factors were identified as significant predictors of domain-specific satisfaction levels. Male gender, higher income status, normal weight, meeting activity guidelines, and healthy dietary habits like sufficient fruit/vegetable intake were all associated with greater satisfaction related to weight, diet, and physical activity. These findings aligned with previous studies and can guide targeted interventions that address specific risk profiles [49]. 

This study highlights the urgent need for comprehensive university health promotion initiatives to counter the escalating obesity and related disease trends compromising young adults’ well-being. The satisfaction/behavior paradox shows that simply providing education, improved resources, and facilities is insufficient. However, this requires participatory interventions tailored to students’ specific readiness to change. Qualitative research is needed to directly assess motivation and habit strength related to the observed risk paradoxes. Multicomponent programs incorporating individual, interpersonal, and environmental levels are also required. By elucidating this complex issue through coordinated initiatives facilitating lifestyle improvements, universities can mitigate both immediate and long-term health consequences threatening this vulnerable cohort. University settings represent an opportune avenue, given their wide reach and formative influence during this critical life stage. Additionally, the controlled university community allows customized interventions to be readily implemented through coordinated efforts.

### Strengths and Limitations

This study has several strengths, including a large sample size of university students, the use of a validated questionnaire assessing lifestyles in a culturally relevant manner, a comprehensive assessment of various lifestyle factors, and the use of robust statistical analyses that were used to examine the associations between these factors and satisfaction levels. Additionally, this study explored the influence of sociodemographic factors, providing valuable insights into potential disparities and the need for tailored interventions. However, the cross-sectional nature of this study precludes the establishment of causal relationships between the variables assessed. Furthermore, the use of self-reported data obtained through direct interviews may have introduced response bias, particularly social desirability bias regarding sensitive topics like dietary choices, substance use, income, and physical activity. This could potentially influence the accuracy of the reported lifestyle behaviors and satisfaction levels. Future self-administered anonymous surveys are needed to validate the findings. Other limitations include the generalizability of this single-university sample. Moreover, cultural and environmental factors may vary across different geographical locations and populations, potentially influencing lifestyle choices and perceptions. More nuanced gender analyses will be undertaken as part of our ongoing work in this area.

## 5. Conclusions

This study reveals high obesity, poor diet, and low activity levels among Saudi university students, representing major public health threats. However, it also uncovers a disconnect between students’ unhealthy lifestyles (especially dietary choices) and elevated satisfaction levels. Minimal dietary satisfaction differences emerged despite participants’ adherence to the guidelines, contrasting with the expected associations between satisfaction and weight/activity status. These findings underscore an urgent need for coordinated university health promotion programs employing motivational interviewing, incremental goal setting, and environmental modifications to raise risk awareness and empower self-efficacy. Implementing such multifaceted initiatives within university settings represents a valuable opportunity, given their wide reach and formative influence during this critical life stage. Additionally, the controlled university community allows for a feasible implementation of customized interventions. Addressing this issue early, before lifestyle patterns become entrenched, can facilitate positive changes, mitigating immediate and lifelong obesity and chronic disease burdens. However, this requires participatory interventions tailored to students’ specific readiness to change. This study suggests that university administrators, health professionals, and policymakers should develop comprehensive health promotion programs for university students. These programs should raise awareness about health risks, motivate change, and provide resources for healthy behaviors. Collaborative partnerships between universities, healthcare providers, and community organizations can ensure sustainability. Future research should examine the temporal relationships between health behaviors, satisfaction levels, and long-term health outcomes. Investing in university students’ well-being can improve population health, productivity, and reduce healthcare costs for future generations. 

## Figures and Tables

**Figure 1 nutrients-16-02246-f001:**
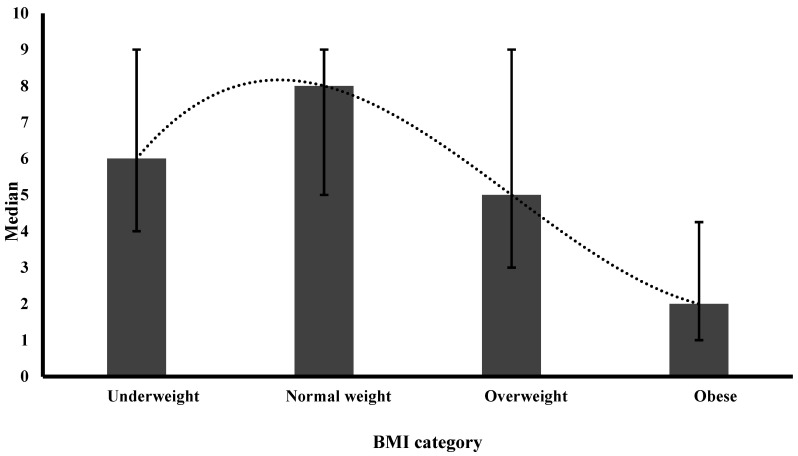
Median satisfaction level of various BMI categories. Error bars represent the interquartile range (IQR); BMI: body mass index; satisfaction was defined by a score of 5 or higher on a 10-point scale, while dissatisfaction was a score below 5.

**Figure 2 nutrients-16-02246-f002:**
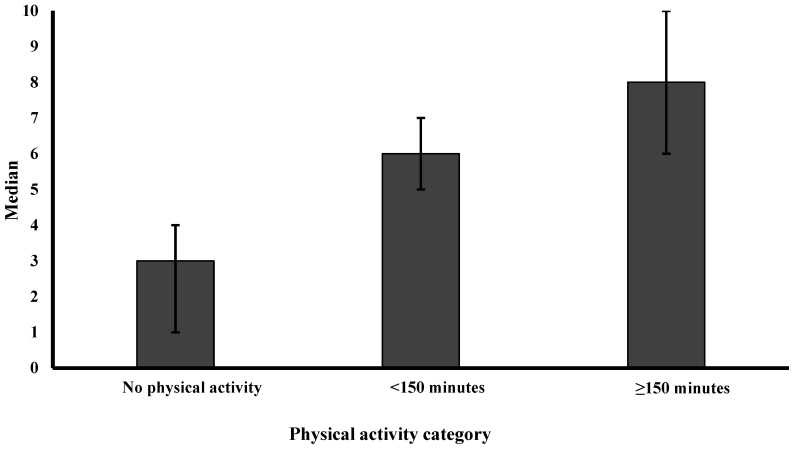
Median satisfaction level of various physical activity levels per week. Error bars represent the interquartile range (IQR); satisfaction was defined by a score of 5 or higher on a 10-point scale, while dissatisfaction was a score below 5.

**Figure 3 nutrients-16-02246-f003:**
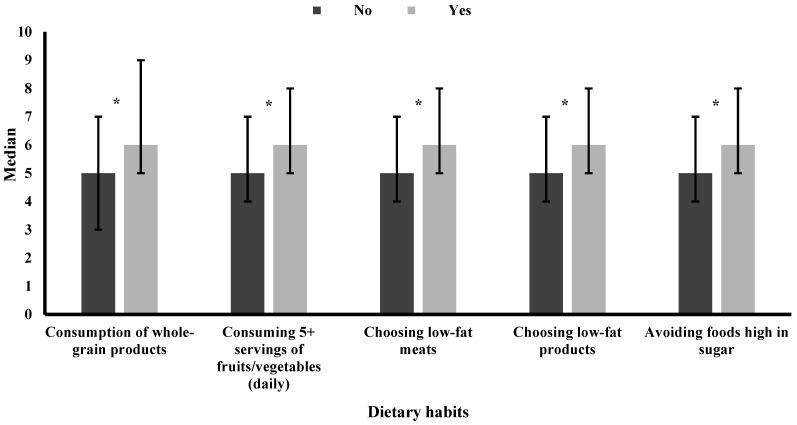
Median satisfaction level of various dietary habits. Error bars represent the interquartile range (IQR); * *p* < 0.05. Satisfaction was defined by a score of 5 or higher on a 10-point scale, while dissatisfaction was a score below 5.

**Table 1 nutrients-16-02246-t001:** Sociodemographic characteristics, self-reported body measurements, and lifestyle behaviors among 1957 university students in Jazan, Saudi Arabia.

Characteristics	Mean ± SD
Age (years)	21.00 ± 1.70
Height (cm) (Total)	164.00 ± 9.30
Male	170.30 ± 6.56
Female	156.40 ± 5.81
Weight (kg) (Total)	60.00 ± 16.00
Male	68.73 ± 16.32
Female	51.42 ± 10.36
BMI (kg/m^2^) ^a^ (Total)	22.00 ± 4.70
Male	23.60 ± 5.02
Female	21.01 ± 4.01
**Characteristics**	**Frequency (%)**
Sex	
Male	1008 (51.51)
Female	949 (48.49)
Residence	
Rural	1011 (51.66)
Urban	946 (48.34)
Family income (monthly, SAR)	
<5000	1207 (61.68)
5000–9999	258 (13.18)
10,000–14,999	236 (12.06)
≥15,000	256 (13.08)
Social status	
Single	1799 (91.93)
Married	144 (7.36)
Divorced/widowed/widower	14 (0.72)
Family housing	
Owned apartment	444 (22.69)
Owned traditional	474 (24.22)
Owned villa	763 (38.99)
Rented	276 (14.10)
Smoking behaviors	
Never smokers	1661 (84.87)
Ex-smokers	62 (3.17)
Current smokers	100 (5.11)
Passive smokers	134 (6.85)
Khat chewing	
Never used it	1880 (96.07)
Ex-user	50 (2.55)
Current user	27 (1.38)
BMI categories	
Underweight	410 (20.95)
Normal weight	1049 (53.60)
Overweight	362 (18.50)
Obese	136 (6.95)
Physical activity (weekly)	
No physical activity	717 (36.64)
<150 min	718 (36.69)
≥150 min	522 (26.67)
Eating behaviors	
Consumption of whole-grain products	1065 (54.42)
Consuming 5 or more servings of fruits/vegetables (daily)	513 (26.21)
Choosing low-fat meats	939 (47.98)
Choosing low-fat products	671 (34.29)
Avoiding foods high in sugar	753 (38.48)

BMI: body mass index; cm: centimeter; kg: kilogram; SAR: Saudi Riyals; SD: standard deviation; ^a^ BMI was calculated using the standard formula of weight in kilograms divided by height in meters squared.

**Table 2 nutrients-16-02246-t002:** Satisfaction levels across BMI categories, physical activity levels, and selected eating behaviors among 1957 university students in Jazan, Saudi Arabia.

Characteristics	Frequency (%)	Median (IQR)
BMI categories ^a^		Weight satisfaction ^#^
Underweight	410 (20.95)	6 (4, 9)
Normal weight	1049 (53.60)	8 (5, 9)
Overweight	362 (18.50)	5 (3, 7)
Obese	136 (6,95)	2 (1, 4.25)
Physical activity (weekly)		Physical activity satisfaction ^#^
No physical activity	717 (36.64)	3 (1, 4)
<150 min	718 (36.69)	6 (5, 7)
≥150 min	522 (26.67)	8 (6, 10)
Eating behaviors		Eating behaviors’ satisfaction ^#^
Consumption of whole-grain products		
Yes	1065 (54.42)	6 (4, 8)
No	892 (45.58)	5 (4, 8)
Consuming 5 or more servings of fruits/vegetables (daily)		
Yes	513 (26.21)	6 (5, 8)
No	1444 (73.79)	5 (4, 7)
Choosing low-fat meats		
Yes	938 (47.93)	6 (5, 8)
No	1019 (52.07)	5 (4, 7)
Choosing low-fat products		
Yes	671 (34.29)	6 (5, 8)
No	1286 (65.71)	5 (4, 7)
Avoiding foods high in sugar		
Yes	753 (38.48)	6 (5, 8)
No	1204 (61.52)	5 (4, 7)

BMI: body mass index; IQR, interquartile range. ^a^ BMI was calculated using the standard formula of weight in kilograms divided by height in meters squared. ^#^ Satisfaction was defined by a score of 5 or higher on a 10-point scale, while dissatisfaction was a score below 5.

**Table 3 nutrients-16-02246-t003:** The distribution of satisfaction level for weight, physical activity, and eating behaviors among 1957 university students in Jazan, Saudi Arabia.

Characteristics	Dissatisfied ^#^ n (%)	Satisfied n (%)
BMI categories ^a^		
Underweight	174 (42.44)	236 (57.56)
Normal weight	268 (25.55)	781 (74.45)
Overweight	198 (54.70)	164 (45.30)
Obese	112 (82.35)	24 (17.65)
Physical activity (weekly)		
No physical activity	631 (88.01)	86 (11,99)
<150 min	321 (44.71)	397 (55.29)
≥150 min	118 (22.61)	404 (77.39)
Eating behaviors		
Consumption of whole-grain products		
Yes	489 (45.92)	576 (54.08)
No	470 (52.69)	422 (47.31)
Consuming 5 or more servings of fruits/vegetables (daily)		
Yes	200 (38.99)	313 (61.01)
No	795 (52.6)	685 (47.4)
Choosing low-fat meats		
Yes	384 (40.94)	554 (59.06)
No	575 (56.43)	444 (43.57)
Choosing low-fat products		
Yes	258 (38.45)	413 (61.55)
No	701 (54.51)	585 (45.49)
Avoiding foods high in sugar		
Yes	303 (40.24)	450 (59.76)
No	656 (54.49)	548 (45.51)

BMI: body mass index; n: sample size. ^a^ BMI was calculated using the standard formula of weight in kilograms divided by height in meters squared. ^#^ Satisfaction was defined by a score of 5 or higher on a 10-point scale, while dissatisfaction was a score below 5.

**Table 4 nutrients-16-02246-t004:** Logistic regression examining predictors of body weight satisfaction, physical activity satisfaction, and eating behaviors satisfaction among 1957 university students in Jazan, Saudi Arabia.

Characteristics	Weight Satisfaction ^#^			Physical Activity Satisfaction ^#^			Eating Behaviors Satisfaction ^#^		
Predictors	OR	95% CI	*p*	OR	95% CI	*p*	OR	95% CI	*p*
Age	1.01	0.95–1.08	0.730	1.09	1.02–1.16	0.014	1.01	0.95–1.08	0.678
Sex [reference: female]									
Male	1.34	1.03–1.74	0.032	1.20	0.90–1.61	0.220	2.25	1.74–2.92	<0.001
Social status [reference: single]									
Married	1.17	0.78–1.76	0.452	0.81	0.52–1.26	0.347	1.13	0.76–1.68	0.543
Family income (monthly, SAR) [reference: <5000]									
5000–9999	0.57	0.41–0.80	0.001	0.77	0.53–1.12	0.170	1.01	0.73–1.39	0.975
10,000–14,999	0.90	0.63–1.27	0.541	0.67	0.45–0.98	0.041	1.12	0.79–1.57	0.520
15,000 and more	1.16	0.83–1.64	0.384	0.90	0.62–1.30	0.568	1.43	1.02–1.99	0.036
Residence [reference: rural]									
Urban	1.27	1.03–1.57	0.026	1.19	0.95–1.50	0.134	1.03	0.84–1.26	0.808
Smoking [reference: never smoked]									
Current smokers	0.80	0.49–1.32	0.374	1.04	0.59–1.83	0.894	0.69	0.43–1.12	0.137
Ex-smokers	0.59	0.33–1.05	0.070	0.56	0.29–1.07	0.079	0.53	0.29–0.94	0.030
Passive smokers	0.76	0.51–1.15	0.189	0.87	0.56–1.36	0.539	0.83	0.56–1.22	0.343
BMI category ^a^ [reference: normal weight]									
Underweight	0.35	0.24–0.51	<0.001	1.01	0.68–1.52	0.943	1.03	0.72–1.47	0.863
Overweight	0.42	0.28–0.64	<0.001	0.59	0.37–0.94	0.026	0.55	0.37–0.83	0.004
Obese	0.17	0.07–0.39	<0.001	0.41	0.16–1.00	0.052	0.33	0.15–0.72	0.006
Physical activity (weekly) [reference: no physical activity]									
<150 min	1.23	0.98–1.55	0.078	8.54	6.49–11.35	<0.001	1.64	1.31–2.06	<0.001
≥150 min	1.56	1.20–2.04	0.001	22.61	16.56–31.21	<0.001	2.02	1.57–2.60	<0.001
Eating Behaviors:									
Consumption of whole grain products	1.15	0.94–1.41	0.175	1.04	0.83–1.29	0.755	1.36	1.12–1.66	0.002
Consuming 5 or more servings of fruits/vegetables (daily)	1.01	0.81–1.28	0.906	1.33	1.04–1.71	0.024	1.58	1.27–1.98	<0.001
Choosing low-fat meats	1.05	0.86–1.29	0.610	1.18	0.95–1.48	0.140	1.61	1.32–1.95	<0.001
Choosing low-fat products	1.23	0.99–1.52	0.066	1.36	1.08–1.71	0.010	1.67	1.36–2.06	<0.001
Avoiding food high in sugar	1.11	0.90–1.37	0.326	1.30	1.04–1.63	0.024	1.58	1.29–1.93	<0.001
R^2^	0.152			0.326			0.140		

BMI: body mass index; SAR: Saudi Riyals. ^a^ BMI was calculated using the standard formula of weight in kilograms divided by height in meters squared. ^#^ Satisfaction was defined by a score of 5 or higher on a 10-point scale, while dissatisfaction was a score below 5.

## Data Availability

The raw data supporting the conclusions of this article are available upon reasonable request to the corresponding author.

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
