# Peer review of "Satisfaction Paradoxes in Health Behaviors: Contrasting Patterns across Weight, Physical Activity and Dietary Habits"

_nutrients, 2024, doi:10.3390/nu16142246_

Round 1

Reviewer 1 Report

Comments and Suggestions for Authors

The authors examine associations between individual characteristics and health behaviours and satisfaction. The manuscript has the potential to add to the literature on understanding factors influencing health behaviours. While the work has numerous merits, including large sample size and thought-through methods, the theoretical and scientific background in the introduction are insufficient. Considering the importance of the satisfaction and later satisfaction paradox, these should be more clearly defined in the introduction.

Authors should ensure that all the relevant background literature used to discuss the results is included in the introduction. This would be especially important as, in the discussion, it is noted that not all the associations were in the expected direction, i.e. even if not stated, the authors had formulated hypotheses. The authors should ensure that relevant studies are included in the introduction and clearly state the hypotheses they tested based on the previous literature. 

Methods would benefit from a better description of the participant recruitment process. While a detailed description of the study is provided elsewhere, it would be beneficial for the reader to have more details about the questionnaires used for the data specific to this manuscript, especially about the questionnaires or questions that evaluated eating and PA behaviours. A minor point, but authors should consider linking the statistical methods to the research hypotheses.

Authors should ensure that results are not duplicated between text and tables or figures. If a result (numerical) is included in the table, this shouldn’t be stated in the text or figures again. For readers unfamiliar with SAR, indicating whether income, e.g. under 5000, is considered low would also be helpful. 

As noted above, the discussion introduces a considerable amount of literature relevant to the introduction. The implicit hypothesis should also be made explicit and discussed explicitly in relation to the results and the previous literature used to formulate the hypotheses. Some of the conclusions, such as “Interventions must address factors perpetuating the risk paradox and employ motivational techniques to modify entrenched habits.” should be checked. This is important considering that it appeared that no data was included about habits or motivation. 

Author Response

Reviewer 1

Thank you for the insightful comments. Here are our point-by-point responses

Comments and Suggestions for Authors

The authors examine associations between individual characteristics and health behaviours and satisfaction. The manuscript has the potential to add to the literature on understanding factors influencing health behaviours. While the work has numerous merits, including large sample size and thought-through methods, the theoretical and scientific background in the introduction are insufficient. Considering the importance of the satisfaction and later satisfaction paradox, these should be more clearly defined in the introduction.

  • Authors should ensure that all the relevant background literature used to discuss the results is included in the introduction. This would be especially important as, in the discussion, it is noted that not all the associations were in the expected direction, i.e. even if not stated, the authors had formulated hypotheses. The authors should ensure that relevant studies are included in the introduction and clearly state the hypotheses they tested based on the previous literature. 

Response:

We appreciate the reviewer highlighting the need to expand the theoretical background in the introduction. In the revised manuscript, we incorporated additional discussion of the satisfaction and satisfaction paradox concepts, including clear definitions and relevant previous literature that informed our hypotheses. We also stated our hypotheses explicitly based on this background.

The revised manuscript states:

In this study, satisfaction refers to an individual's overall contentment and positive assessment of their current lifestyle behaviors, specifically their body weight status, level of physical activity engagement, and dietary habits. A growing body of research highlights concerning rates of obesity, inactivity, and poor diets among university students in SA and neighboring countries [10-13]. However, associations between health behaviors and related satisfaction remain unclear. Some studies reveal coexistence of obesity and related risks with high levels of satisfaction that can be referred to as “risk paradox”. In contrast, other evidence indicates lower satisfaction related to inactivity or excess weight [14-16]. Promoting nutritional change also faces unique challenges compared to increasing exercise, with taste preferences, ingrained habits, and low self-efficacy hampering dietary improvements despite motivations to eat healthier [17-19].

Given these conflicting findings and the concept of the "satisfaction paradox" where individuals exhibit risks like obesity yet report high satisfaction, the current study aimed to elucidate relationships between lifestyle factors and satisfaction levels among Saudi university students. We hypothesized that students exhibiting obesity, physical inactivity, and poor dietary habits would report lower satisfaction compared to students with healthier lifestyles. Recognizing these risks and perceptions will inform tailored interventions to mitigate obesity and chronic disease burdens impacting Saudi youth.

  • Methods would benefit from a better description of the participant recruitment process. While a detailed description of the study is provided elsewhere, it would be beneficial for the reader to have more details about the questionnaires used for the data specific to this manuscript, especially about the questionnaires or questions that evaluated eating and PA behaviours. A minor point, but authors should consider linking the statistical methods to the research hypotheses.

Response:

We agree that more details on participant recruitment and the specific questionnaires used would strengthen the methods. In the revised manuscript, we provided further information on the recruitment process, questionnaires administered, and details on the key measures evaluated related to eating, physical activity, and other health behaviors. We also linked the statistical analyses section more closely to the hypotheses.

The revised manuscript states the following added sentences:

Convenience sampling was used to recruit study participants from various university colleges. To ensure representative sampling, we selected a proportional number of students from the chosen colleges.

Recruitment involved identification and direct approach of targeted students on university campus. Students who agreed to participate completed the interview.

The questionnaire captured information on demographics, lifestyle factors, dietary habits, physical activity, tobacco and khat use, nutrition, weight, and height. Assessed variables aligned with Saudi guidelines on the prevention and management of obesity [34]. Self-reported physical activity was evaluated by asking if participants met the recommended 150 minutes per week threshold. Dietary behaviors were assessed through questions on adherence to healthy eating plate recommendations, specifically, consumption of whole-grain products, fruits and vegetables, low-fat meats, and low-fat products, and avoidance of high sugar foods. The students were also asked to rate their overall satisfaction with their lifestyle.

Continuous numerical variables with a normal distribution were reported as means and standard deviations (SD), while variables with skewed distribution were presented as medians and interquartile range (IQR).

Median satisfaction and IQRs were compared across BMI categories, physical activity levels, and adherence to healthy eating recommendations. For binary analyses, a cutoff of 5 was used to categorize satisfaction levels. Participants with scores of 5 or higher were classified as satisfied, while those below 5 were classified as dissatisfied.

Students' satisfaction with weight, physical activity levels, and dietary behaviors were examined as outcome variables.

  • Authors should ensure that results are not duplicated between text and tables or figures. If a result (numerical) is included in the table, this shouldn’t be stated in the text or figures again. For readers unfamiliar with SAR, indicating whether income, e.g. under 5000, is considered low would also be helpful. 

Response:

Thank you for catching the potential duplication between tables/figures and text. We thoroughly reviewed the results to ensure data is not redundantly presented and have removed any duplications. Most of the text in the results section narrating tables modified completely and highlighted in red. The new text is as follows:

“This cross-sectional study included 1,957 Saudi university students with a mean age of 21 years. As shown in Table 1, the sample contained nearly equal proportions of males and females. Height and weight differed by gender, with greater values among males resulting in higher average BMIs compared to females. Place of residence was also evenly split between rural and urban areas. Over half had relatively low family incomes, were single, and lived in owned housing. Rates of smoking and khat use were low overall. Regarding BMI status, over half were normal weight, while around 25% were overweight/obese. Many participants had insufficient physical activity, with only about 27% meeting guidelines. Around half exhibited certain healthy eating habits. The proportion who reported engaging in specific healthy dietary habits ranged from about 26% to 54%.

Table 2 displays the median and IQR of satisfaction levels across BMI categories, physical activity levels, and selected eating behaviors. Satisfaction levels demonstrated a reverse J-shaped pattern across BMI categories, with the highest satisfaction among normal weight and the lowest satisfaction among obese individuals (Figure 1). For physical activity, satisfaction showed a dose-response relationship (Figure 2), with the highest satisfaction reported among participants achieving the recommended 150 minutes or more per week. Those engaging in less than 150 minutes per week had lower satisfaction, while participants reporting no activity had the lowest satisfaction. Minimal differences were observed in satisfaction levels between individuals adhering to healthy eating behaviors compared to those who did not (Figure 3).

Table 3 shows the distribution of satisfaction levels for body weight, physical activity, and eating behaviors. Satisfaction was defined by a score of 5 or higher on a 10-point scale, while dissatisfaction was a score below 5. Satisfaction levels varied noticeably across different BMI categories, with the highest dissatisfaction among obese participants and the highest satisfaction among normal weight participants. For physical activity, individuals engaging in no activity predominantly expressed dissatisfaction, while those engaging in 150 minutes or more per week demonstrated higher satisfaction. Minimal differences were observed in satisfaction levels among individuals practicing healthy eating habits compared to those who did not.

The multivariate analysis reveals the predictors associated with satisfaction levels regarding weight, physical activity, and eating behaviors. As demonstrated in Table 4, factors like older age, male gender, higher income, urban residence, normal BMI, meeting physical activity guidelines, and healthy eating habits were associated with greater satisfaction in one or more domains. Conversely, overweight/obesity tended to predict lower satisfaction.”

For income levels, we thank the reviewer for the suggestion to better contextualize the income levels reported in SAR. In the revised manuscript, we have added the following statement when describing Table 1:

"Over half had relatively low family incomes”

  • As noted above, the discussion introduces a considerable amount of literature relevant to the introduction. The implicit hypothesis should also be made explicit and discussed explicitly in relation to the results and the previous literature used to formulate the hypotheses.

Response:

We appreciate the reviewer's suggestion to more clearly link the discussion and interpretation of our results back to the original hypotheses stated in the introduction. As recommended, we have revised the discussion to explicitly restate our hypothesis that students with obesity, inactivity, and poor diets would have lower satisfaction compared to those with healthier lifestyles. The added text notes this was based on previous literature demonstrating correlations between unhealthy behaviors and lower satisfaction. We agree this provides helpful context for readers on the conceptual foundation and expected relationships that motivated the study aims and analyses.

The revised discussion now includes the following introductory statement:

“This study aimed to assess lifestyle choices and relevant satisfaction among university students in Jazan, Saudi Arabia. We hypothesized that students exhibiting obesity, physical inactivity, and poor dietary habits would report lower satisfaction levels compared to students with healthier lifestyles. However, the current findings revealed a more complex relationship.”

  • Some of the conclusions, such as “Interventions must address factors perpetuating the risk paradox and employ motivational techniques to modify entrenched habits.” should be checked. This is important considering that it appeared that no data was included about habits or motivation. 

Response:

We appreciate the reviewer's comment that our conclusion about addressing motivational factors and entrenched habits went beyond the direct data collected. This is a fair critique. To clarify, motivation and habit strength were not directly measured by our survey. We agree that the phrasing implies our data assessed these factors, which was not the case here. Our conclusion that interventions should employ motivational techniques and aim to modify habits was based on interpreting the satisfaction-behavior paradox findings, rather than direct measurements. However, we agree this inference should be checked, as highlighted by the helpful reviewer comment.

In the revised manuscript, we have modified the conclusion to state:

"However, this requires participatory interventions tailored to students' specific readiness to change. Qualitative research is needed to directly assess motivation and habit strength related to the observed risk paradoxes."

Thank you again for your constructive feedback.

References

  1. Moradi-Lakeh, M.; El Bcheraoui, C.; Afshin, A.; Daoud, F.; AlMazroa, M.A.; Al Saeedi, M.; Basulaiman, M.; Memish, Z.A.; Al Rabeeah, A.A.; Mokdad, A.H. Diet in Saudi Arabia: findings from a nationally representative survey. Public Health Nutrition 2017, 20, 1075-1081, doi:10.1017/S1368980016003141.
  2. Althumiri, N.A.; Basyouni, M.H.; AlMousa, N.; AlJuwaysim, M.F.; Almubark, R.A.; BinDhim, N.F.; Alkhamaali, Z.; Alqahtani, S.A. Obesity in Saudi Arabia in 2020: Prevalence, Distribution, and Its Current Association with Various Health Conditions. Healthcare (Basel) 2021, 9, doi:10.3390/healthcare9030311.
  3. Ministry of Health, K.o.S.A. World Health Survey - Saudi Arabia 2019 Final Report. 2019; p 194.
  4. Al-Jawaldeh, A.; Abbass, M.M. Unhealthy dietary habits and obesity: the major risk factors beyond non-communicable diseases in the eastern mediterranean region. Frontiers in nutrition 2022, 9, 817808.
  5. Kuk, J.L.; Ardern, C.I.; Church, T.S.; Hebert, J.R.; Sui, X.; Blair, S.N. Ideal weight and weight satisfaction: association with health practices. American journal of epidemiology 2009, 170, 456-463.
  6. Tyrovolas, S.; Koyanagi, A.; Stickley, A.; Haro, J.M. Weight perception, satisfaction, control, and low energy dietary reporting in the US adult population: results from the National Health and Nutrition Examination Survey 2007-2012. Journal of the Academy of Nutrition and Dietetics 2016, 116, 579-589.
  7. Schlomann, A.; Seifert, A.; Rietz, C. Relevance of activity tracking with mobile devices in the relationship between physical activity levels and satisfaction with physical fitness in older adults: representative survey. JMIR aging 2019, 2, e12303.
  8. Vosgerau, J.; Scopelliti, I.; Huh, Y.E. Exerting self‐control≠ sacrificing pleasure. Journal of Consumer Psychology 2020, 30, 181-200.
  9. Gardner, B.; Arden, M.A.; Brown, D.; Eves, F.F.; Green, J.; Hamilton, K.; Hankonen, N.; Inauen, J.; Keller, J.; Kwasnicka, D., et al. Developing habit-based health behaviour change interventions: twenty-one questions to guide future research. Psychology & Health 2023, 38, 518-540, doi:10.1080/08870446.2021.2003362.
  10. Adriaanse, M.A.; Vinkers, C.D.W.; De Ridder, D.T.D.; Hox, J.J.; De Wit, J.B.F. Do implementation intentions help to eat a healthy diet? A systematic review and meta-analysis of the empirical evidence. Appetite 2011, 56, 183-193, doi:https://doi.org/10.1016/j.appet.2010.10.012.
  11. Kingdom of Saudi Arabia, M.o.H. Saudi guidelines on the prevention and management of obesity. 1st Edition.; 2016.

Reviewer 2 Report

Comments and Suggestions for Authors

This is a quite interesting study regarding the paradox of satisfaction despite inappropriate dietary choices and limited physical activity among students of one of the universities in Saudi Arabia. Introduction justifies the study. Results are well discussed. Conclusions summarize the study where authors indicate future needs.

I have serious doubts about the methods. The data were obtained through a direct interview; therefore, the question arises how reliable the answers regarding inappropriate dietary choices, addictions, income, and physical activity are. The results could have looked completely different if the survey had been conducted anonymously. This should definitely be pointed out as a limitation of this study.

I completely don't understand why did you calculate the average height, weight and BMI for both sexes, you have enough participants to do it separately.

Figures 1a-c should be independent figures, and by the way, why not to calculate significance of the differences for figure 1C?

Besides:

Your abstract suggests that participants calculated their own BMI

The phrase “150+ minutes” should be expressed better, it currently resembles the artificial language introduced by George Orwell in his novel 1984

Author Response

Reviewer 2

Thank you for the insightful comments. Here are our point-by-point responses

Comments and Suggestions for Authors

This is a quite interesting study regarding the paradox of satisfaction despite inappropriate dietary choices and limited physical activity among students of one of the universities in Saudi Arabia. Introduction justifies the study. Results are well discussed. Conclusions summarize the study where authors indicate future needs.

  • I have serious doubts about the methods. The data were obtained through a direct interview; therefore, the question arises how reliable the answers regarding inappropriate dietary choices, addictions, income, and physical activity are. The results could have looked completely different if the survey had been conducted anonymously. This should definitely be pointed out as a limitation of this study.

Response:

We appreciate the reviewer raising this important limitation regarding potential response bias in our direct interview methods. As suggested, we have expanded the limitations section to highlight that the face-to-face compared to self-administered anonymous surveys format may have impacted the accuracy of responses on sensitive topics like diet, activity, and other behaviors. The revised discussion now includes the following:

“Furthermore, the use of self-reported data obtained through direct interviews may have introduced response bias, particularly social desirability bias regarding sensitive topics like dietary choices, substance use, income, and physical activity. This could potentially influence the accuracy of the reported lifestyle behaviors and satisfaction levels. Future self-administered anonymous surveys are needed to validate the findings.”

  • I completely don't understand why did you calculate the average height, weight and BMI for both sexes, you have enough participants to do it separately.

Response:

The reviewer makes a good observation. Calculating BMI averages for men and women separately would better account for gender differences. In the revised manuscript, we added the average height, weight and BMI for both sexes and added these in table 1 and updated the relevant results.

  • Figures 1a-c should be independent figures, and by the way, why not to calculate significance of the differences for figure 1C?

Response:

Thank you for catching this oversight. We agree that presenting the figures independently will improve clarity. We updated the figures to be separate. Adding statistical tests of differences for Figure 3 is also a helpful suggestion that we implemented in the revised manuscript.

  • Besides, your abstract suggests that participants calculated their own BMI

Response: We thank the reviewer for catching the confusing statement in the abstract regarding BMI calculation. As noted, participants did not calculate their own BMIs. We have revised the abstract and deleted “BMI” from the following sentence

“Methods: In this cross-sectional study, 1,957 students at Jazan University completed surveys on demographics, body mass index (BMI), physical activity, dietary habits, and 10-point satisfaction scales for weight, activity, and diet.”

Additionally, we have revised the methods section to state:

"BMI was calculated based on measured height and weight using the standard formula of weight in kilograms divided by height in meters squared."

  • The phrase “150+ minutes” should be expressed better, it currently resembles the artificial language introduced by George Orwell in his novel 1984

Response: We agree the phrase "150+ minutes" can be expressed better. In the revision, we modified it to more clearly as “150 minutes or more”

Thank you again for this constructive feedback.

Reviewer 3 Report

Comments and Suggestions for Authors

First, I would like to thank the journal editor for the opportunity to review this paper.

As the authors indicate, Saudi Arabia has very high obesity rates, which undoubtedly impact the population's health. Therefore, the paper's topic is justified and interesting.

Line 42-43: In line 41, the authors state that regular physical activity should be performed and cite reference (10). Then, in line 43, they indicate the number of minutes and cite a reference from the AHA. These recommendations are outdated. I suggest they update the bibliographic reference to the WHO's 2020 physical activity recommendations, which recommend increasing activity levels to 300 minutes per week.

A key point of this study is to inform how the different variables were measured.

In section 2.4. Study variables and measures (line 105), the authors should clearly indicate how the various variables were assessed. For example: How were physical activity levels evaluated? Was it with the IPAQ? I request the authors provide this information. Without it, it is impossible to conduct a thorough review of the paper. They must clearly state how all the variables were evaluated.

In the paper, the authors include, in my opinion, too many variables.

I suggest they focus on the most important ones: BMI, diet, and physical activity levels (if possible, in minutes). Aspects such as being divorced or widowed are, in my opinion, irrelevant, and the data confirm this with 14 (0.72) in Table 1. Therefore, I recommend that, after analysis, they select the relevant variables.

Additionally, the sample is sufficiently balanced in terms of gender: Male (51.51%). Therefore, I recommend that they create a Table with disaggregated data for males and females separately, as gender is a very relevant variable.

I recommend reviewing the IQR data (Table 2). The IQR is a value, not a range.

Similarly, for Table 1, in Table 3, information should be disaggregated by sex (male vs female).

In Table 4, I recommend the authors review the R-square value and include significant variables in the logistic regression analysis. As I mentioned earlier (e.g., the variable marital status (divorced) does not make sense to analyse.

In summary: to recommend the publication of this paper, I need to know how the variables were evaluated. Without this information, it is not possible for me to review the paper.

Author Response

First, I would like to thank the journal editor for the opportunity to review this paper.

As the authors indicate, Saudi Arabia has very high obesity rates, which undoubtedly impact the population's health. Therefore, the paper's topic is justified and interesting.

  1. Line 42-43: In line 41, the authors state that regular physical activity should be performed and cite reference (10). Then, in line 43, they indicate the number of minutes and cite a reference from the AHA. These recommendations are outdated. I suggest they update the bibliographic reference to the WHO's 2020 physical activity recommendations, which recommend increasing activity levels to 300 minutes per week.

Response:

Thank you for pointing this out. We have updated the reference to the WHO's 2020 physical activity recommendations. The new bibliographic reference is number 20

  1. A key point of this study is to inform how the different variables were measured.

In section 2.4. Study variables and measures (line 105), the authors should clearly indicate how the various variables were assessed. For example: How were physical activity levels evaluated? Was it with the IPAQ? I request the authors provide this information. Without it, it is impossible to conduct a thorough review of the paper. They must clearly state how all the variables were evaluated.

Response:

We appreciate you raising this point. In the revised manuscript, we have provided more explicit details in the Methods Section how the various variables were assessed, including:

  • Physical activity levels were self-reported using a validated questionnaire.
  • Dietary behaviors were assessed through questions on adherence to healthy eating plate recommendations
  • BMI was calculated based on measured height and weight using the standard formula of weight in kilograms divided by height in meters squared.
  • The students were also asked to rate their overall satisfaction with their lifestyle. Satisfaction was defined by a score of 5 or higher on a 10-point scale, while dissatisfaction was a score below 5

The revised manuscript states the following added sentences:

Convenience sampling was used to recruit study participants from various university colleges. To ensure representative sampling, we selected a proportional number of students from the chosen colleges.

Recruitment involved identification and direct approach of targeted students on university campus. Students who agreed to participate completed the interview.

The questionnaire captured information on demographics, lifestyle factors, dietary habits, physical activity, tobacco and khat use, nutrition, weight, and height. Assessed variables aligned with Saudi guidelines on the prevention and management of obesity [34]. Self-reported physical activity was evaluated by asking if participants met the recommended 150 minutes per week threshold. Dietary behaviors were assessed through questions on adherence to healthy eating plate recommendations, specifically, consumption of whole-grain products, fruits and vegetables, low-fat meats, and low-fat products, and avoidance of high sugar foods. The students were also asked to rate their overall satisfaction with their lifestyle.

Continuous numerical variables with a normal distribution were reported as means and standard deviations (SD), while variables with skewed distribution were presented as medians and interquartile range (IQR).

Median satisfaction and IQRs were compared across BMI categories, physical activity levels, and adherence to healthy eating recommendations. For binary analyses, a cutoff of 5 was used to categorize satisfaction levels. Participants with scores of 5 or higher were classified as satisfied, while those below 5 were classified as dissatisfied.

Students' satisfaction with weight, physical activity levels, and dietary behaviors were examined as outcome variables.

  1. In the paper, the authors include, in my opinion, too many variables.

I suggest they focus on the most important ones: BMI, diet, and physical activity levels (if possible, in minutes). Aspects such as being divorced or widowed are, in my opinion, irrelevant, and the data confirm this with 14 (0.72) in Table 1. Therefore, I recommend that, after analysis, they select the relevant variables.

Response:

We appreciate this valuable suggestion. In the revised manuscript, we have prioritized the main variables of interest - BMI, dietary habits, and physical activity levels in discussion. Additionally, we have removed the "Divorced/Widow/Widower" category from the "Social status" variable. Additionally, we removed the "Khat chewing" and “Family housing” variables from all three models in table 4.

  1. Additionally, the sample is sufficiently balanced in terms of gender: Male (51.51%). Therefore, I recommend that they create a Table with disaggregated data for males and females separately, as gender is a very relevant variable.

Response:

We appreciate the reviewer's suggestion to provide a gender-disaggregated analysis. While an in-depth examination of gender differences was not a central aim of this particular study, we acknowledge the importance of exploring such variations. As mentioned, this research is part of a larger research project to comprehensively evaluate the health needs of the Jazan University community. Subsequent papers stemming from this broader project will delve deeper into potential gender-based analyses.

To address the reviewer's feedback, we have added a note in the limitations section clarifying that more nuanced gender analyses will be undertaken as part of our ongoing work in this area. Additionally, in the revised manuscript, we have included the average height, weight, and BMI values for both male and female participants. These data have been incorporated into Table 1, and the relevant results sections have been updated accordingly.

  1. I recommend reviewing the IQR data (Table 2). The IQR is a value, not a range.

Response:

Thank you for catching this. We have revised Table 2 to report the IQR as values rather than ranges.

  1. Similarly, for Table 1, in Table 3, information should be disaggregated by sex (male vs female).

Response:

As pointed out in response to point 4, we appreciate the reviewer's suggestion to provide a gender-disaggregated analysis. This research is part of a broader research project that will explore gender differences in more depth in subsequent work stemming from this project, as noted in the revised limitations section.

  1. In Table 4, I recommend the authors review the R-square value and include significant variables in the logistic regression analysis. As I mentioned earlier (e.g., the variable marital status (divorced) does not make sense to analyse.

Response:

We have reviewed the R-square value and provided them for the different models in table 4. Also, we have removed the "Divorced/Widow/Widower" category from the "Social status" variable due to the small number of cases in that group and the lack of significant associations observed. Additionally, we have dropped the "Khat chewing" and “Family housing” variables from all three models as it did not significantly predict any of the outcome variables (weight satisfaction, physical activity satisfaction, or eating behavior satisfaction).

  1. In summary: to recommend the publication of this paper, I need to know how the variables were evaluated. Without this information, it is not possible for me to review the paper.

Response:

As mentioned in point 2, we have now provided clear details in the Methods Section on how each of the key variables like physical activity, diet, and BMI were specifically measured and evaluated in this study.

Thank you again for your constructive feedback.

Round 2

Reviewer 1 Report

Comments and Suggestions for Authors

I would like to thank authors for their thoughtful consideration of the feedback. No further comments.

Reviewer 2 Report

Comments and Suggestions for Authors

I accept the current version of the paper

Reviewer 3 Report

Comments and Suggestions for Authors

The authors have addressed the issues raised in my previous report